# Genetics of Functional Seizures; A Scoping Systematic Review

**DOI:** 10.3390/genes14081537

**Published:** 2023-07-27

**Authors:** Ali A. Asadi-Pooya, Mark Hallett, Nafiseh Mirzaei Damabi, Khatereh Fazelian Dehkordi

**Affiliations:** 1Epilepsy Research Center, Shiraz University of Medical Sciences, Shiraz 71438, Iran; mirzaie.n59@gmail.com (N.M.D.); kh.fazelian@gmail.com (K.F.D.); 2Jefferson Comprehensive Epilepsy Center, Department of Neurology, Thomas Jefferson University, Philadelphia, PA 19107, USA; 3Division of Intramural Research, National Institute of Neurological Disorders and Stroke (NINDS), NIH, Bethesda, MD 20892, USA; hallettm@ninds.nih.gov

**Keywords:** gene, dissociative, polymorphism, psychogenic, seizure

## Abstract

Background: Evidence on the genetics of functional seizures is scarce, and the purpose of the current scoping systematic review is to examine the existing evidence and propose how to advance the field. Methods: Web of science and MEDLINE were searched, from their initiation until May 2023. The following key words were used: functional neurological disorder(s), psychogenic neurological disorder(s), functional movement disorder(s), psychogenic movement disorder(s), functional seizures(s), psychogenic seizure(s), nonepileptic seizure(s), dissociative seizure(s), or psychogenic nonepileptic seizure(s), AND, gene, genetic(s), polymorphism, genome, epigenetics, copy number variant, copy number variation(s), whole exome sequencing, or next-generation sequencing. Results: We identified three original studies. In one study, the authors observed that six (5.9%) patients with functional seizures carried pathogenic/likely pathogenic variants. In another study, the authors observed that, in functional seizures, there was a significant correlation with genes that are over-represented in adrenergic, serotonergic, oxytocin, opioid, and GABA receptor signaling pathways. In the third study, the authors observed that patients with functional seizures, as well as patients with depression, had significantly different genotypes in *FKBP5* single nucleotide polymorphisms compared with controls. Conclusion: Future genetic investigations of patients with functional seizures would increase our understanding of the pathophysiological and neurobiological problems underlying this common neuropsychological stress-associated condition.

## 1. Introduction

Functional neurological disorders (FND) reflect impairments in brain networks, leading to motor, sensory, or cognitive signs and symptoms; they are common in neurology clinics. The four variants of FND, i.e., functional seizures (FS), functional movement disorders (FMD), persistent perceptual postural dizziness (PPPD), and functional cognitive disorder, show similarities in etiology and pathophysiology [1]. The pathophysiology of FND includes over-activity of the limbic system and dysfunction of the brain networks that are involved in cognitive and movement processes [1]. In particular, abnormal functional connectivity between emotion processing areas of the brain with regions associated with executive control and cognitive processes, as well as the functional connections of the anterior cingulate cortex, may be important in the neurobiology and pathophysiology of functional seizures [2,3,4]. Furthermore, a few recent studies have provided evidence for altered structural brain connectivity in patients with functional seizures [2]. Similarly, in FMD, brain connectivity studies suggest aberrant connections between the amygdala and motor areas, temporo-parietal junction, and insula [5]. 

In spite of the above statements, currently, we do not know the precise neurobiological underpinnings and pathophysiological mechanisms of functional seizures; as a result, it is very hard to design targeted clinical trials to develop effective methods of treatment for patients with functional seizures [2]. A previous meta-analysis that used data from 13 manuscripts on 228 patients with functional seizures showed that 47% of patients achieved seizure control after completion of a psychological therapeutic intervention [6]. However, a study of the natural history of 69 patients with functional seizures, who had never received proper psychological treatment, showed that 52% of the patients were seizure-free at their last follow-up visit [7]. Furthermore, a study of 368 patients with functional seizures (CODES trial) showed that psychotherapy (i.e., cognitive behavioral therapy) in addition to routine care had no statistically significant benefit compared with routine care alone for the reduction of the frequency of functional seizures [8]. The scientific community should invest more time into providing a clear understanding of the pathophysiology and neurobiology of functional seizures, to develop more effective methods of treatment for these patients [2]. 

Having said that, converging evidence shows that brain connectivity is under genetic influence [9]. Considering the evidence on functional and structural brain connectivity differences in patients with FND, including those with functional seizures (compared with healthy controls), it is plausible to assume that various FND, and functional seizures in particular, are under genetic influence, at least to some extent. It seems that anatomical brain connectivity is under a more powerful genetic influence than functional brain connectivity, and these genetic influences are not consistently distributed over the brain [9]. Therefore, it is reasonable to hypothesize that phenotypic variations in certain regions and connections are under stronger genetic control than others, and it is plausible to assume that various FND (e.g., FMD vs. functional seizures) are under certain different genetic influences, as well as certain common influences. 

Evidence on the genetics of FND in general, and functional seizures in particular, is scarce, and the purpose of the current endeavor was to examine the existing evidence and propose how to advance the field in a productive way. Scoping reviews are a useful tool in evidence-synthesis approaches. Scoping reviews may be preferable over systematic reviews where the purpose is to identify knowledge gaps and how research is conducted on a certain topic, among other indications [10]. The focus of this endeavor is on functional seizures.

Functional (psychogenic) seizures are typically characterized by sudden and paroxysmal clinical events that semiologically may resemble epileptic seizures but that are not due to underlying epileptic activity; ictal electroencephalography (EEG) does not show any abnormal discharges [11,12]. “Functional seizures” are commonly encountered in neurology and epilepsy clinics around the world. A systematic review and an analytical study estimated that the incidence of functional seizures was 3.1 (95% Confidence Interval CI: 1.1–5.1) per 100,000 population per year, and the prevalence rate of functional seizures in 2019 was 108.5 (95% CI: 39.2–177.8) per 100,000 population, in the USA [13]. Patients with functional seizures are at higher risk of having psychiatric comorbidities compared with the general population; this is comparable with patients with epilepsy [11]. In addition, functional seizures often have crippling effects on patients’ lives. Furthermore, the evidence suggests that patients with functional seizures have an increased rate of mortality compared to those in the general population; this increased rate of mortality is comparable to that among patients with epilepsy [11].

## 2. Materials and Methods

MEDLINE (accessed from PubMed) and Web of science, from their initiation until May 2023, were systematically searched for published articles about the topic of interest. In these two electronic databases, the following search strategy was executed and the following key words were used (in title): functional neurological disorder(s), psychogenic neurological disorder(s), functional movement disorder(s), psychogenic movement disorder(s), functional seizures(s), psychogenic seizure(s), nonepileptic seizure(s), dissociative seizure(s), or psychogenic nonepileptic seizure(s), AND, gene, genetic(s), polymorphism, genome, epigenetics, copy number variant, copy number variation(s), whole exome sequencing, or next-generation sequencing. The inclusion criteria were all human studies on the genetics of functional seizures (i.e., retrospective, cross sectional, case-control, case series, etc.) and articles written in English. The exclusion criteria were non-original studies (i.e., reviews, corresponding letters, etc.). 

Two authors (NM and KF) scanned the reference lists of the obtained studies and previous reviews to add any relevant publications. The authors gained access to the full reports for all the manuscripts that met the inclusion and exclusion criteria or where there was any uncertainty. They resolved any disagreements through discussions with the first author. The following data were extracted from the included publications: the first author, country and year of the publication, study methodology, main results, and main limitations. This was a qualitative (descriptive) study and there was no statistical analysis. The level of evidence of the included studies was determined following https://onlinelibrary.wiley.com/pb-assets/assets/23788038/Levels_of_Evidence-1519834967260.pdf (accessed on 17 December 2022) [14].

## 3. Results

We identified three original studies on the genetics of functional seizures (Figure 1 and Table 1) [15,16,17]. All three studies provided a low level of evidence. However, considering the significance of these three studies, we have summarized their findings and conclusions below:(a)The first study, which was conducted by Costin Leu and colleagues, investigated the neurological disorder-associated genetic variants in patients with functional seizures [15]. The authors generated whole-exome sequencing and whole-genome genotyping data to identify rare pathogenic (P) or likely pathogenic (LP) variants in 102 patients with functional seizures and 448 patients with epilepsy. Variants were classified for all patients based on the recommendations of the American College of Medical Genetics and Genomics and the Association for Molecular Pathology guidelines [18]. The authors considered genes that are associated with neurological or psychiatric disorders as candidate genes for functional seizures (a limitation of this study). They observed that six (5.9%) patients with functional seizures (without comorbid epilepsy) carried pathogenic/likely pathogenic variants (deletions at 10q11.22-q11.23, 10q23.1-q23.2, distal 16p11.2, and 17p13.3, and nonsynonymous variants in *NSD1* and *GABRA5*) [15]. However, the burden of P/LP variants among patients with functional seizures was alike to the burden found in patients with epilepsy. The four identified deletions in patients with functional seizures in their study had previously been reported in epilepsy and other neurological disorders (with high phenotypic variability and incomplete penetrance) [15]. The *NSD1* gene product (enzyme) controls the activity of genes that are involved in normal growth and development [19]. *GABRA5* (γ-Aminobutyric Acid Type A Receptor Subunit Alpha5) influences inhibitory activity; so far, diseases that have been associated with this gene include epileptic encephalopathies [20]. Costin Leu and colleagues concluded that it is likely that these genetic aberrations impair neurodevelopmental processes in a nonspecific way and, therefore, contribute to the genetic variance of a broad spectrum of brain disorders. The specific disease phenotype (e.g., functional seizures) is probably further specified by the interplay with genetic background effects and environmental factors [15].(b)The second study, which was performed by Johannes Jungilligens and colleagues, investigated spatial similarities between imaging-derived phenotypes and Allen human brain atlas (AHBA) gene expression profiles, with an interest in identifying genetic pathways that are dually suggested in association of volumetric gray matter variations with symptom severity and trauma burden (in 20 adult patients with functional seizures) [16]. They used self-report questionnaires (Somatoform Dissociation Questionnaire-20 and Traumatic Experiences Checklist). They also used voxel-based morphometry preprocessing of magnetic resonance imaging (MRI). Finally, to explore potential relationships between gray matter and gene expression profiles, they used a data-driven approach, utilizing spatial similarity metrics between gray matter statistical maps and regional gene expression patterns (AHBA) [16]. They observed that, in patients with functional seizures at the intersection of SDQ-20 (symptom severity) and sexual trauma imaging-derived phenotypes, there was significant spatial correlation with genes that are over-represented in adrenergic, serotonergic, oxytocin, opioid, and GABA receptor signaling pathways [10]. The authors concluded that adverse life experiences and symptom severity were associated with gray matter volumes in cingulo-insular and amygdala areas, spatially overlapping with the expression patterns of genes that are involved in stress-related signaling and neurodevelopment processes [16]. However, they postulated that it is not likely that these gene variations are functional seizure-specific; these may represent genetic susceptibilities that, combined with adverse life events, may lead to functional seizures or other FND phenotypes [16].(c)The third study was conducted by our team [17]. *FKBP5* is a co-chaperone of hsp90 that regulates glucocorticoid receptor sensitivity [21]. *FKBP5* single-nucleotide polymorphisms (SNPs) have been associated with an increased risk of different psychiatric disorders (e.g., depression and post-traumatic stress disorder (PTSD)) in previous studies [22,23,24,25,26]. Furthermore, interactions between the *FKBP5* gene and early-life traumatic experiences (e.g., childhood sexual trauma) may increase the likelihood of stress-related disorders later in the life [25]. In this research, the authors investigated whether there were associations between two common *FKBP5* polymorphisms (rs9470080 and rs1360780) and functional seizures in a case-control study. Seventy patients with functional seizures, 140 with major depressive disorder (MDD), and 140 healthy controls (HC) were studied. They observed that patients with functional seizures and those with depression had significantly different genotypes in both SNPs compared with those in the HC group. However, the authors could not exclude the potential confounding effects of depression [17]. They concluded that “Further genetic investigations of patients with functional seizures may increase our understanding of the neurobiological underpinnings of this condition, but such studies should be large enough and very well-designed; they should include a comparison group with depression (and probably, PTSD or anxiety) in addition to a healthy control group” [17].

## 4. Discussion

We have provided a brief summary of the results from three studies that investigated the genetic underpinnings of functional seizures. The sample sizes/methodologies of the existing publications are such that these studies are largely hypothesis generating/exploratory. Furthermore, the results of the studies have not been validated elsewhere, and robust evidence on the genetic etiology of functional seizures is so far lacking in the literature. However, the hypothesis of such an etiology for functional seizures is very plausible based on the emerging evidence on the modified functional and structural brain connectivity patterns in patients with functional seizures [2]. In addition, there is plenty of evidence in the literature on the genetic basis of other stress-associated neuropsychological disorders and some evidence for the genetic basis of other FND. A systematic review provided strong evidence of interactions between *FKBP5* genotypes and early-life stress, which could pose a significant risk for stress-associated neuropsychological disorders (e.g., depression and PTSD) [25]. Previous studies have also implicated a strong genetic architecture for anxiety disorders [27]. One study suggested that hypermethylation of a discrete region located within the *SLC6A4* promoter region in women could underlie differential serotonin transporter (SERT) expression in women compared with in men; this could be one of the underlying pathological underpinnings through which women show increased prevalence of somatization [28]. Another study of 69 patients with FMD showed that the tryptophan hydroxylase 2 (*TPH2*) gene polymorphism may modulate functional movement disorders, both directly and interactively with childhood trauma [29]. *TPH2* has its most significant impact in the serotonin pathway. The study of somatization in other conditions often also points to serotonin [30,31,32]. A study showed increased methylation of the oxytocin receptor gene in 15 patients with motor functional neurological disorder compared with healthy controls [33]. Finally, a neuroimaging-gene expression study of 30 patients with motor FND implicated the role of genes [34], as shown in Table 2. Table 2 shows the original studies on the genetics of FND that were identified in the current study (while this was not the focus of the current endeavor) [29,33,34].

Functional seizures, FMD, and other stress-associated disorders share a common core of manifestations and clinical characteristics; therefore, it is reasonable to investigate whether variants in stress-related genes (e.g., glucocorticoid and serotonin receptor signaling pathways) also contribute to the development of various FNDs, including functional seizures. However, it is not likely that any of these gene variations are specific to functional seizures or other FNDs; they may represent genetic susceptibilities that, combined with life stressors and other biological and environmental factors, may lead to individual FND phenotypes (e.g., through epigenetics). While a candidate gene approach limits the likelihood of a discovery, a well-selected gene panel (that included glucocorticoid, serotonin, and oxytocin receptor signaling pathways for the purpose of this study) can investigate pathological processes in modestly sized cohorts, by focusing on a limited number of pathways with greater neurobiological relevance [29]. In addition to the recommendation to continue studying candidate genes related to stress/mood/attachment-based systems, it would be informative if future consideration was also given to candidate genes in other domains of motor and sensory/awareness processing, to continue pushing the field’s conceptualization of functional seizures. All of the studies conducted so far have been small, but there are hints in several of them that there may well be genetic underlying pathomechanisms for various stress-associated neuropsychological conditions, including functional seizures. Future genetic investigations of patients with functional seizures may advance our understanding of the underlying pathomechanisms and the neurobiological underpinnings of this common neuropsychological stress-associated condition, but such studies should be large and well-designed and include a comparison group with depression, in addition to a healthy control group.

## Figures and Tables

**Figure 1 genes-14-01537-f001:**
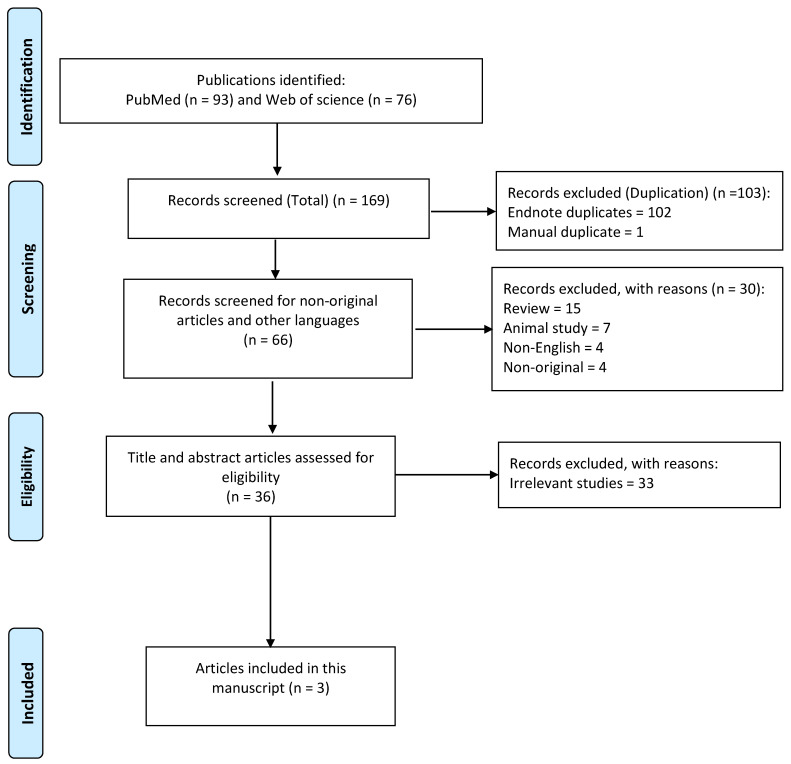
The PRISMA (preferred reporting items for systematic reviews and meta-analyses) flow diagram of the study.

**Table 1 genes-14-01537-t001:** Original studies on the genetics of functional seizures.

Author/Year/ Country	Methods	Main Results	Level of Evidence	Limitations
Leu, C./2020/USA	Whole-exome sequencing and whole-genome genotyping to identify rare, pathogenic (P) or likely pathogenic (LP) variants in 102 patients with functional seizures and 448 individuals with epilepsy.	Six (5.9%) patients with functional seizures (only) had P/LP variants. The burden of P/LP types among people with functional seizures was similar to the burden observed in people with epilepsy.	3b (Individual case-control study)	Psychiatric comorbidities
Jungilligens, J./2022/Netherlands	Questionnaires, structural MRIs, and Allen human brain atlas gene expression information were used to probe the intersection of symptom severity, adverse life experiences burden, and gray matter volumes in 20 patients with functional seizures.	Adverse life experiences and symptom severity were associated with gray matter volumes in cingulo-insular and amygdala areas, spatially overlapping with expression patterns of genes involved in stress-related signaling and neurodevelopment.	4 (Case series without comparison)	Small sample size, psychiatric comorbidities
Asadi-Pooya, AA/2023/Iran	Seventy patients with functional seizures, 140 with depression (MDD), and 140 healthy controls were studied. Their DNAs were analyzed for the rs1360780 in the 3′ region and rs9470080 in the 5′ region of the *FKBP5*.	Patients with functional seizures and those with MDD had less GG and more AA genotypes in both rs9470080 and rs1360780 SNPs compared with those in healthy controls. There were no significant differences between functional seizures and MDD groups in terms of genotype frequencies for both SNPs.	3b (Individual case-control study)	Tested only two SNPs within FKBP5

**Table 2 genes-14-01537-t002:** Original studies on the genetics of functional neurological disorders (FND).

Author/Year/Country	Methods	Main Results	Level of Evidence	Limitations
Apazoglou, K./2018/Switzerland	Epigenetic changes in the promoter of the oxytocin receptor gene (*OXTR*) between 15 patients with motor FND and 16 HC were explored.	Significantly higher levels of methylation of the *OXTR* gene was found in patients compared with that in controls (68.1 ± 4.3 vs. 62.5 ± 6.8, *p* = 0.01).	4 (Case series with comparison)	Small sample size, psychiatric comorbidities
Spagnolo, P./2020/USA	A total 69 patients with FMD were genotyped for 18 SNPs from 14 candidate genes. Resting-state functional connectivity data were obtained in a subgroup of 38 patients with FMD and 38 HC.	A tryptophan hydroxylase 2 (*TPH2*) gene polymorphism-G703T-significantly predicted clinical and neurocircuitry manifestations of FMD. The *TPH2* genotype showed a significant interaction with childhood trauma in predicting worse symptom severity.	3b (Individual case-control study)	Small sample size, not a genome-wide approach
Diez, I./2021/USA	A neuroimaging-gene expression study. Effects of early-life maltreatment on resting-state functional connectivity architecture in 30 patients with motor FND were assayed. Then, they compared trauma endophenotypes in FND with regional-differences in transcriptional gene expression as measured by the AHBA.	Physical abuse correlated connectivity maps overlapped with the AHBA spatial expression of three gene-clusters: (i) neuronal morphogenesis and synaptic transmission genes in limbic and paralimbic areas; (ii) locomotory behavior and neuronal generation genes in left-lateralized structures; and (iii) nervous system development and cell motility genes in right-lateralized structures.	4 (Case series with comparison)	Small sample size, psychiatric comorbidities, and phenotypic heterogeneity

FMD: functional motor disorder; AHBA: Allen human brain atlas; HC: healthy control; SNP: single nucleotide polymorphism; FND: Functional neurological disorders.

## Data Availability

Not applicable.

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
