# Peer review of "Genetics of Functional Seizures; A Scoping Systematic Review"

_genes, 2023, doi:10.3390/genes14081537_

Round 1

Reviewer 1 Report

Thank you so much for giving me the opportunity to review this important and interesting paper. I think it is very well organized and written and it is concerning an important topic of genetic underprints of the functional neurological disorders. The small amount of trials that are finally included in this paper supports the hypothesis that there is a need for further investigation of this topic. I have only some small suggestions that the authors might consider to introduce to improve this paper. Maybe it would be valuable to compare with other functional neurologic symptoms and their genetic underprints? I know that the authors mention this briefly in the discussion, but maybe it could be extended. 

Author Response

Response.  Thanks for this thoughtful comment. There are 3 studies on the genetics of FND in the literature. We discussed these in the text and in Table 2.

In addition, there is plenty of evidence in the literature on the genetic basis of other stress-associated neuropsychological disorders and some evidence on the genetic basis of other FNDs. A systematic review provided strong evidence of interactions between FKBP5 genotypes and early-life stress that could pose a significant risk for stress-associated disorders (e.g., MDD and PTSD) [19]. Previous studies have also implicated a strong genetic architecture of anxiety disorders [21]. One study suggested that hypermethylation of a discrete region located within SLC6A4 promoter region in women could underlie differential serotonin transporter (SERT) expression in women compared with that in men; this could be one of the underlying pathological mechanisms by which women exhibit increased prevalence of somatization [22]. Another study of 69 patients with functional movement disorders showed that the tryptophan hydroxylase 2 (TPH2) gene polymorphism may modulate FMD both directly and interactively with childhood trauma [23]. TPH2 has its most significant impact in the serotonin pathway. Study of somatization in other conditions also often point to serotonin [24-26]. A study showed increased methylation of the oxytocin receptor gene in 15 patients with motor functional neurological disorder compared with healthy controls [27]. Finally, a neuroimaging-gene expression study of 30 patients with motor FND implicated the role of genes as shown in the Table 2.

Reviewer 2 Report

The present investigation is a scoping systematic review of studies investigating genetic underpinnings of functional seizures (FS), a type of functional neurological disorder. Albeit limited in the possibilities for conclusions to date, this review provides an important call to action for more studies aiming to understand genetic contributions to FS.

The search terms appear adequate to have captured studies relevant to the genetics of FS. The review focused on three studies that fit the search criteria. This approach of going into greater detail about these three studies was a good alternative, given the limitations in terms of quantity of available studies. A strength of the present review was its discussion of limitations (as well as potential) of the studies reviewed. Conclusions are minimally informative in terms of identifying genetic associations specific to FS versus epilepsy, psychopathology (e.g., MDD), or early life stress more generally. However, this again underscores the need for more work in this area, and the overlap is telling as well.

The figure and tables are effective in illustrating the review process and methods/findings of the target studies. Level of evidence definitions could be noted very briefly along with their numbers (e.g., 3b Individual Case-control study), or in the Table note. (The link to the cited article for levels of evidence was not working in the manuscript, although presumably would be in a published version.) I did wonder what kinds of studies were the 33 with relevant search terms but ultimately excluded; they may have been completely irrelevant, but if there was a pattern (e.g., studies of other biological processes in FS, genetics of other dissociative processes), it may be useful to note.

It was not clear from the description in the Discussion if there were many more studies on the genetics of non-FS FND (e.g., motor FND), and only the three in Table 2 were highlighted given their citation in Apazoglou et al. (ref 27) -- or if there were very few studies on the genetics of non-FS FND as well. As noted, an exhaustive review of the genetics of FMD was beyond the scope of the present focus on FS, but the state of the literature could be characterized or contrasted a bit more.

The conclusion is consistent with evidence to date (e.g., from functional connectivity and other studies, as well as from the very few presented here) that it may be unlikely to find candidate genes with one-to-one mappings specific to FS (as is true of psychopathology in general). In addition to the recommendation to continue studying candidate genes related to stress/mood/attachment-based systems, it would be informative if future consideration is given to candidate genes in other domains of motor and sensory/awareness processing, to continue pushing the field’s conceptualization of FS.

In sum, functional neurological disorders, including FS, are prevalent in neurology clinics and emergency departments. Given the history whereby these conditions were (and still are) described as “psychological” (conversion, hysteria), the fact that there is a biological basis that can be subject to rigorous empirical investigation has been relatively overlooked until recent years. As such, there has been minimal systematic study of the genetic basis of FND, particularly FS. The present investigative review, therefore, is an important advance. Language use: Overall clear and well written. There are a couple of very minor errors (e.g., missing words). “Irrelative” in Fig. 1 sounds a bit awkward versus “Irrelevant” – but perhaps this was a deliberate word choice.

Author Response

Reviewer 2.

Level of evidence definitions could be noted very briefly along with their numbers (e.g., 3b Individual Case-control study), or in the Table note.

Response. Done as suggested in the Table.

I did wonder what kinds of studies were the 33 with relevant search terms but ultimately excluded; they may have been completely irrelevant, but if there was a pattern (e.g., studies of other biological processes in FS, genetics of other dissociative processes), it may be useful to note.

Response. They were really irrelevant as noted in the figure. Some examples include: PMID: 31739099, PMID: 35306367, PMID: 31252271, PMID: 30982134, PMID: 30959273, PMID: 27164717, PMID: 26956567.

It was not clear from the description in the Discussion if there were many more studies on the genetics of non-FS FND (e.g., motor FND), and only the three in Table 2 were highlighted given their citation in Apazoglou et al. (ref 27) -- or if there were very few studies on the genetics of non-FS FND as well. As noted, an exhaustive review of the genetics of FMD was beyond the scope of the present focus on FS, but the state of the literature could be characterized or contrasted a bit more.

Response. We cannot answer this question as this was not the focus of our study. We identified only 3 studies on other FND and we included all of them in the text and Table 2. To answer this question a separate study on all subtypes of FND is required (e.g., tremor, movement, cognitive disorder, etc.).

The conclusion is consistent with evidence to date (e.g., from functional connectivity and other studies, as well as from the very few presented here) that it may be unlikely to find candidate genes with one-to-one mappings specific to FS (as is true of psychopathology in general). In addition to the recommendation to continue studying candidate genes related to stress/mood/attachment-based systems, it would be informative if future consideration is given to candidate genes in other domains of motor and sensory/awareness processing, to continue pushing the field’s conceptualization of FS.

Response. Revised as suggested.

In sum, functional neurological disorders, including FS, are prevalent in neurology clinics and emergency departments. Given the history whereby these conditions were (and still are) described as “psychological” (conversion, hysteria), the fact that there is a biological basis that can be subject to rigorous empirical investigation has been relatively overlooked until recent years. As such, there has been minimal systematic study of the genetic basis of FND, particularly FS. The present investigative review, therefore, is an important advance. Language use: Overall clear and well written. There are a couple of very minor errors (e.g., missing words). “Irrelative” in Fig. 1 sounds a bit awkward versus “Irrelevant” – but perhaps this was a deliberate word choice.

Response. Revised. Dear Editor, please replace Fig 1 with the revised one that is attached, thanks.
